# Efficiency of InN/InGaN/GaN Intermediate-Band Solar Cell under the Effects of Hydrostatic Pressure, In-Compositions, Built-in-Electric Field, Confinement, and Thickness

**DOI:** 10.3390/nano14010104

**Published:** 2024-01-01

**Authors:** Hassan Abboudi, Haddou EL Ghazi, Redouane En-nadir, Mohamed A. Basyooni-M. Kabatas, Anouar Jorio, Izeddine Zorkani

**Affiliations:** 1LPS, Faculty of Sciences, Mohamed Ben Abdellah University, Fes 30000, Morocco; 22SMPI Group, ENSAM Laboratory, Hassan II University, Nile 150, Casablanca 20670, Morocco; 3Dynamics of Micro and Nano Systems Group, Department of Precision and Microsystems Engineering, Delft University of Technology, Mekelweg 2, 2628 CD Delft, The Netherlands; 4Solar Research Laboratory, Solar and Space Research Department, National Research Institute of Astronomy and Geophysics, Cairo 11421, Egypt; 5Department of Nanotechnology and Advanced Materials, Graduate School of Applied and Natural Science, Selçuk University, 42030 Konya, Turkey

**Keywords:** IBSC, III-nitrides, efficiency, semi-graded potential, built-in field, thickness

## Abstract

This paper presents a thorough numerical investigation focused on optimizing the efficiency of quantum-well intermediate-band solar cells (QW-IBSCs) based on III-nitride materials. The optimization strategy encompasses manipulating confinement potential energy, controlling hydrostatic pressure, adjusting compositions, and varying thickness. The built-in electric fields in (In, Ga)N alloys and heavy-hole levels are considered to enhance the results’ accuracy. The finite element method (FEM) and Python 3.8 are employed to numerically solve the Schrödinger equation within the effective mass theory framework. This study reveals that meticulous design can achieve a theoretical photovoltaic efficiency of quantum-well intermediate-band solar cells (QW-IBSCs) that surpasses the Shockley–Queisser limit. Moreover, reducing the thickness of the layers enhances the light-absorbing capacity and, therefore, contributes to efficiency improvement. Additionally, the shape of the confinement potential significantly influences the device’s performance. This work is critical for society, as it represents a significant advancement in sustainable energy solutions, holding the promise of enhancing both the efficiency and accessibility of solar power generation. Consequently, this research stands at the forefront of innovation, offering a tangible and impactful contribution toward a greener and more sustainable energy future.

## 1. Introduction

In the realm of solar energy conversion, intermediate-band solar cells (IBSCs) stand out as a captivating focus within the scientific community. Propelled by an innovative energy conversion mechanism, IBSCs transcend the boundaries of conventional photovoltaic approaches. The fusion of quantum phenomena and materials science is reshaping the landscape of solar energy utilization, pushing the boundaries of classical efficiency and spectral limits. Inspired by principles in quantum engineering and materials science, our exploration into IBSCs aims to unveil the untapped potential of solar energy. By capturing a broader spectrum of photons, our research enhances electric current generation, achieving efficiencies that surpass the traditional Shockley–Queisser limit. This marks a pivotal step towards revolutionizing solar energy utilization [1,2]. This class of solar cells has demonstrated significant promise by effectively transforming low-energy photons into electric power [3]. According to the National Renewable Energy Laboratory (NREL), IBSC photovoltaic cells achieve the highest efficiency under experimental conditions (47.1%) [4]. This solar cell category relies on intermediate bands (IBs) achieved through QWs positioned within the material’s bandgap, which allow for the absorption of sub-bandgap energies. The inclusion of these IBs enables significantly improved photon absorption and electron generation processes, leading to a remarkable enhancement in power conversion efficiency [5]. The IBs’ inclusion significantly boosts photon absorption, resulting in a higher output current. These nanostructured mini-bands surpass the Shockley–Queisser limit that balances photogeneration and radiative recombination, thereby exceeding efficiency boundaries and providing cost-effective photovoltaic solutions. In the radiative limit, IBSCs achieve an efficiency of 63.2%, surpassing single-gap (40.7%) and two-junction (55.4%) solar cells at their radiative limits [6]. To surpass the constraints set by the Shockley–Queisser threshold for solar cell efficiency, researchers have proposed several methods. One involves boosting photon absorption through the optical plasmonic effect, while simultaneously diminishing the exciton binding energy using the inner electrical plasmonic effect, with a specific focus on perovskite solar cells. This innovative approach, distinct from conventional p–n junction cells, opens up new avenues and holds the potential for substantial improvements in solar cell efficiency [7]. In the study by M. Laska et al., the investigation into the influence of plasmonic nanoparticles on internal cell electricity revealed that these nanoparticles enhance solar cell efficiency. This enhancement was attributed to the plasmon-mediated photovoltaic effect observed in both semiconductor- and chemical-type solar cells. The effect goes beyond merely increasing sunlight absorption [8,9]. In 2000, Antonio Martí and his research group from the Political University of Madrid—Instituto de Energies Solaire, Madrid, Spain discussed the possibility of fabricating the intermediate-band solar cell (IBSC), a cell with the potential to achieve 63.2% efficiency under concentrated sunlight, using quantum dot (QD) technology [10]. The Stranski–Krastanov technique is suggested for achieving this aim [11]. Quantum dots (QDs) undergo a unique tensile-strained self-assembly on surfaces that are lattice-matched to material substrates, deviating unexpectedly from the traditional Stranski–Krastanov (SK) growth mode. Unlike the typical SK growth, where quantum dots form on a fixed wetting-layer (WL) thickness, they can be manufactured using several methods [12,13]. However, in 2005, the same research group introduced a device with conversion efficiency surpassing the 40.7% limit value observed in InAs-based QD single-gap cells. This achievement was substantiated through electroluminescence and quantum efficiency measurements [14].

To date, a very limited number of studies have delved into intermediate-band solar cells (IBSCs), spanning both the theoretical and experimental realms. Theoretical and numerical explorations have primarily centered around single intermediate-band solar cells (SIBSCs), employing diverse nanostructures such as InAsN/AlPSb [15], InAs/InGaAs quantum dots (QDs) [16], and InN/GaN quantum wells (QWs) [17]. In III-nitride semiconductor devices grown along the wurtzite c-axis, the absence of inversion symmetry results in non-zero spontaneous and piezoelectric polarizations directed out of the plane. GaN and $In_{x}Ga_{1-x}N$ wurtzite structures, as evidenced in prior research, exhibit substantial inherent macroscopic polarization [18]. Prior studies unveiled significant inherent macroscopic polarization in wurtzite GaN and In_x_Ga_1−x_N structures. However, due to substantial lattice mismatch, both between InN and GaN (11%) and between GaN and In_x_Ga_1−x_N, a substantial bias field arises in [0001]-oriented InGaN quantum wells grown on thick GaN. Consequently, a robust integrated built-in electric field (BEF) of around MV/cm and potent piezoelectric polarization emerges [15,16]. In this computational study, our objective was to explore IBSCs fabricated from (In, Ga)N semiconductor materials, given their capacity to effectively absorb the vast majority of the visible electromagnetic spectrum because of the adjustable bandgap of its alloys, especially $In_{x}Ga_{1-x}N$. Wurtzite (WZ) semiconductor heterostructures, including $In_{x}Ga_{1-x}N/GaN$ QW, composed of wide-bandgap group III-nitrides, garner notable interest for their potential in electronics, optoelectronics, and photovoltaics. The thin (In, Ga)N active layer contributes to high quantum efficiency, a pivotal feature for these applications. $In_{x}Ga_{1-x}N$, a ternary compound, exhibits outstanding traits—encompassing the solar spectrum (0.78 eV to 3.42 eV), high absorption, radiation resistance, thermal stability, and exceptional chemical robustness [19,20]. Moreover, the optical properties of InGaN/GaN systems under different internal and external conditions have been intensively investigated [21,22,23,24,25,26]. Optoelectronic devices employing these materials can experience degradation due to the strong integrated electric field (BEF), leading to a decline in overall performance. To address this issue, we opted for a wurtzite InGaN semi-graded quantum well (SGQW) structure. This choice maintains growth similarities with conventional structures, thereby minimizing the increase in crystalline defects. Moreover, the material responses undergo changes under hydrostatic pressure, affecting crucial characteristics such as effective masses, dielectric constants, bandgap energy, and lattice constants. Therefore, a comprehensive understanding of both internal and external factors that influence the solar cells’ performance becomes essential for optimizing their efficiency. Several studies underscore the significance of this consideration, highlighting the influence of piezoelectric polarization on the efficiency of Ga-face GaN/InGaN solar cells. This emphasizes the importance of a holistic approach in addressing various factors to enhance the overall efficiency of optoelectronic devices [27]. Based on their simulations, polarization charges detrimentally impact solar cells’ performance. This stems from the elevated energy barrier height for holes and the induced electric field at the GaN/InGaN interface, hampering efficient carrier collection. Consequently, both short-circuit current and open-circuit voltage decline considerably. In their work, R. Belghouthi et al. [28] introduced a basic analytical model addressing polarization’s effects on InGaN double-heterojunction solar cells. Concerning pressure’s impact on solar cells’ performance, Oyelade et al. explored its effects on perovskite solar cells’ photoconversion efficiency. Their findings revealed a decline in photoconversion efficiency as the pressure increased [29]. El Aouami et al. recently examined how the internal electric field, arising from polarization within the active region of the p-i-n photodiode, affects the characteristics of InN/InGaN quantum-dot (QD) intermediate-band solar cells. Nevertheless, their study overlooked both the polar nature of (In, Ga)N-based nanolayers and the stress induced by hydrostatic pressure [30]. The optical properties of InGaN/GaN quantum wells (QWs) are crucial in the field of optoelectronics, especially for applications such as solar cells, detectors, light-emitting diodes (LEDs), and laser diodes [31]. Composed of alternating layers of InGaN and GaN, InGaN/GaN QWs allow for wavelength tuning by adjusting the indium composition, spanning from ultraviolet to visible wavelengths [32]. These structures exhibit high luminescence efficiency attributed to the quantum confinement effect within the wells, enhancing radiative recombination. Additionally, carrier localization and the quantum-confined Stark effect (QCSE) influence optical properties by shifting energy levels and modifying the emission spectrum. Minimizing spectral line broadening, influenced by factors like indium composition fluctuations and interface roughness, is essential for applications requiring narrow linewidths. Temperature dependence affects carrier dynamics, and achieving optical gain in InGaN/GaN QWs is crucial for their use in semiconductor lasers [33,34]. Ongoing research actively addresses challenges such as the “green gap” and material degradation, focusing on new designs and manufacturing techniques to enhance the efficiency and reliability of InGaN/GaN QWs in practical optoelectronic devices [35].

This study investigates factors affecting the efficiency of semi-graded quantum-well intermediate-band solar cells (SGQW-IBSCs), including BEF, hydrostatic pressure, chemical composition, confinement, layer thickness, donor impurities, and heavy-hole levels. To counter BEF effects, we use a semi-graded structure with varied chemical compositions. Employing the finite element method (FEM), we numerically solve the Schrödinger equation within the framework of the effective-mass approximation. Our research addresses gaps in exploring the interplay of hydrostatic pressure, BEF, impurities, size, confinement, and indium composition on IBSC efficiency with a semi-graded GaN/InN/InGaN/GaN (SGQW-IBSC) system. This paper includes an introduction, theoretical framework, discussion, and conclusion. This work deepens our understanding, paving the way for advancements in intermediate-band solar cell technology.

## 2. Theory and Models

### 2.1. Energy Levels and Electronic States

Our research focuses on examining the effects of hydrostatic pressure, impurity locations, size variations, and compositions (indium: In) on the performance of intermediate-band solar cells (IBSCs) made out of wurtzite (WZ) Ga-faced GaN/InN/(In, Ga)N/GaN strained semi-graded quantum wells (SGQWs) grown along the [0001] direction (*c*-axis) of bulk material. We take into consideration both spontaneous polarization and piezoelectricity in our analysis to create a more accurate representation of the system, given the prevalent polar nature of III-nitride semiconductors in most cases. In the SGQW configuration, there are two interconnected quantum wells (QWs): one is formed using InN (referred to as the left QW or LQW) with a thickness of $l_{1}$, and the other is constructed using InGaN (referred to as the right QW or RQW) with a thickness of $l_{2}$. These QWs are enclosed by external GaN barriers, each with a thickness denoted as L. This structure is incorporated within the intrinsic region (I-region) of the device. The *c*-axis of the WZ structure is aligned parallel to the growth direction (*z*-axis). Figure 1a displays a comprehensive schematic of a GaN/InN/InGaN/GaN-based IBSC, showcasing the active region, contacts, and substrate. In Figure 1b, the illustration depicts potential profiles, energy levels, and wave functions for ground- and excited-state electrons in the conduction band (CB) and holes in the valence band (VB) without external excitation.

The energy levels and their corresponding wave functions are determined through the solution of the one-dimensional time-independent Schrödinger equation, expressed as follows:(1)$$H\psi \left(z\right)=E\psi \left(z\right)$$

The Hamiltonian system governing the behavior of a single particle (i.e., electron, hole) within the studied system is defined as presented in Equation (2). This equation considers factors such as a mobile hydrogen-like impurity, composition, pressure, and the built-in electric field (BEF) effect. This equation is calculated within the framework of both the one-band parabolic theory and the effective mass approach. It has been numerically solved using the finite element method (FEM) due to the increased complexity arising from the inclusion of the Coulombian term (impurity), rendering it almost analytically unsolvable [36,37,38].
(2)$$H=-\frac{\hbar^{2}}{2}\vec{\nabla }\left(\frac{1}{m_{i}^{*}\left(x,p,z\right)}\vec{\nabla }\right)\psi_{i}\left(z\right)-\frac{\alpha_{i}{\;e}^{2}}{\varepsilon^{*}\left(x,p,z\right)\varepsilon_{0}\left|{\vec{r}}_{i}-{\vec{r}}_{0}\right|}\psi_{i}\left(z\right)+V_{T}\left(x,p,z\right)\psi_{i}\left(z\right)=E_{i}\psi_{i}\left(z\right)\;\left(i=e,h\right)$$
where e is the electron charge and $\alpha_{e(h)}=1(-1)$, $\left|{\vec{r}}_{i}-{\vec{r}}_{0}\right|$ denotes the electron–impurity distance, while $\varepsilon_{0}$ represents the dielectric constant of the vacuum; $m_{i}^{*}\left(x,p,z\right)$ and $\varepsilon^{*}\left(x,p,z\right)$ are the electron’s effective mass and the relative dielectric constant, respectively, both contingent upon the pressure, composition, and displacement of the particle.

$V_{T}\left(x,p,z\right)$ is the pressure- and position-dependent total potential energy, expressed as follows:(3)$$V_{T}\left(x,p,z\right)=V_{i}\left(x,p,z\right)+\Phi \left(x,p,z\right)$$

The first term $V_{i}\left(x,p,z\right)$ is the electron(hole) confinement potential due to the band offset in the WZ GaN/InN/InGaN/GaN SGQW, expressed as in Ref. [39]:(4)$$V_{i}\left(x,p,z\right)=Q_{i}∆{E_{g}}_{j}^{\Gamma }$$
with
(5)$$∆{E_{g}}_{j}^{\Gamma }=∆{E_{g}}_{j}^{\Gamma }\left(unstrained\right)+\delta {E_{c}}_{j}^{strain}$$

The band-offset ratio, defined as the ratio between the conduction band offset ($\Delta E_{c}$) and the valence band offset ($\Delta E_{v}$), is assumed to be $Q_{i}=0.63\left(0.37\right)(i\equiv e,h)$, where $∆{E_{g}}_{j}^{\Gamma }\left(unstrained\right)$ is the bandgap energy between the barrier and the well at point Γ. The second term $\Phi \left(x,p,z\right)$ is the static electric potential. Taking into account the lattice mismatch between the LQW, RQW, and barrier, the static electric potential term, denoted as Φ(x,p,z), encompasses the influence of the internal electric field on the polarization charges [39].
(6)$$\delta {E_{c}}_{j}^{strain}=2a_{c}\left(\frac{a^{w}}{a^{b}}-1\right)\left(1-\frac{c_{12}^{w}}{c_{11}^{w}}\right)$$
with
(7)$$\left\{\begin{array}{c} j=1,\;\;\;\;\;\;\;\;\;\;\;\;\;\;\;\;\;For\;\;\;\;\;\;\;\;\;\;\;0\;\leq z\leq L\\ j=2,\;\;\;\;\;\;\;\;\;\;\;\;\;\;\;\;\;\;\;\;For\;\;\;\;\;\;\;\;L\;\leq z\leq L+l_{1}\\ j=3,\;\;\;\;\;\;\;\;\;\;\;For\;\;\;\;\;\;L+l_{1}\;\leq z\leq L+l_{1}+l_{2}\\ j=4,\;\;\;\;\;\;\;\;\;For\;\;\;\;\;\;\;L+l_{1}+l_{2}\leq z\leq 2L+l_{1}+l_{2} \end{array}\;\;\;\;\;\;\;\;\;\;\;\;\;\;\right.$$

### 2.2. Built-In Electric Field

According to Equation (3), the second term is given as follows:(8)$$\Phi \left(x,p,z\right)=\alpha eF_{j}\left(x,p,z\right)z\;$$
where $F_{j}$ represents the BEF in various regions of our structure, attributable to piezoelectricity and spontaneous polarizations. The orientation-dependent $F_{j}$ electric field arises from the interplay of piezoelectricity, spontaneous polarization, crystal polarity, and strains within the SGQW system. The alignment of both piezoelectricity and spontaneous polarization coincides with the z-direction. As a result, the effect of the BEF aligns with the growth direction, as depicted in Figure 1a [40].

For simplification, we will adopt the approach of Takeuchi et al. [41] and disregard the intricate strains arising in the GaN layers due to the lattice and thermal mismatch between GaN and the substrate. Furthermore, we will not account for the alterations in the lattice constant of the thin strained ${In}_{x}{Ga}_{1-x}N$ layer caused by biaxial compressive stress [42], allowing us to obtain
(9)$$2LF_{GaN}+l_{1}F_{InN}+l_{2}F_{InGaN}=0$$

Over this computation, our initial consideration involves a thin layer of InN wells that have been meticulously grown atop a substantial layer of GaN. Subsequently, a slender layer of InGaN wells is coherently developed atop a thick InN layer along the [0001] direction. This is followed by a GaN barrier layer. The strain within the InGaN layer varies according to the distinct indium composition and diverse growth conditions. Moreover, it is crucial to note that the formula for the strength of $F_{j}$ due to the BEF is derived based on the assumption that the potential energies at the extreme left and right of our SGQW structure are equal. Furthermore, by expanding the first-order polarization of the integrated electric field and applying boundary conditions that ensure the continuity of the electric displacement vector at the hetero-interfaces within our SGQW structure, we can obtain the following expressions [43,44]:(10)$$\left\{\begin{array}{c} \varepsilon_{InN}\varepsilon_{0}F_{InN}=\varepsilon_{GaN}\varepsilon_{0}F_{GaN}+P_{GaN}-P_{InN}\\ \varepsilon_{w}\varepsilon_{0}F_{w}=\varepsilon_{InN}\varepsilon_{0}F_{InN}+P_{InN}-P_{w} \end{array}\right.$$

The size-dependent behavior of the built-in electric field (BEF) for various regions within the studied system (SG-DW) is formulated as follows [27]:(11)$$F_{j}=\left\{\begin{array}{c} F_{1}=F_{GaN}=\frac{\left(l_{1}\varepsilon_{w}P_{InN}+l_{2}\varepsilon_{InN}P_{w}\right)-P_{GaN}\left(l_{1}\varepsilon_{w}+l_{2}\varepsilon_{InN}\right)}{\varepsilon_{0}\left[2L\varepsilon_{InN}\varepsilon_{w}+l_{1}{\varepsilon_{GaN}\varepsilon }_{w}+l_{2}{\varepsilon_{GaN}\varepsilon }_{InN}\right]}\;\;\;\;\;\;\;\;\;\;0\;\leq z\leq L\;\\ F_{2}=F_{InN}=\frac{\left(2L\varepsilon_{w}P_{GaN}+l_{2}\varepsilon_{GaN}P_{w}\right)-P_{InN}\left(2L\varepsilon_{w}+l_{2}\varepsilon_{GaN}\right)}{\varepsilon_{0}\left[2L\varepsilon_{InN}\varepsilon_{w}+l_{1}{\varepsilon_{GaN}\varepsilon }_{w}+l_{2}{\varepsilon_{GaN}\varepsilon }_{InN}\right]}\;\;\;\;\;\;\;\;\;L\;\leq z\leq L+{l\;}_{1}\\ F_{3}=F_{w}=\frac{\left(2L\varepsilon_{w}P_{GaN}+l_{2}\varepsilon_{GaN}P_{InN}\right)-P_{w}\left(2L\varepsilon_{InN}+l_{1}\varepsilon_{GaN}\right)}{\varepsilon_{0}\left[2L\varepsilon_{InN}\varepsilon_{w}+l_{1}{\varepsilon_{GaN}\varepsilon }_{w}+l_{2}{\varepsilon_{GaN}\varepsilon }_{InN}\right]}\;\;\;\;L+{l\;}_{1}\;\leq z\leq L+{l\;}_{1}+{l\;}_{2}\\ F_{4}=0\;\;\;\;\;\;\;\;\;\;\;\;\;\;\;\;\;\;\;\;\;\;\;\;\;\;\;\;\;\;\;\;\;\;\;\;\;\;\;\;\;\;\;\;\;\;\;\;\;\;\;\;\;\;\;\;\;\;\;\;\;\;\;\;\;\;\;\;\;\;\;\;\;\;\;\;\;\;\;\;\;\;\;\;\;z\geq L+{l\;}_{1}+{l\;}_{2} \end{array}\right.$$
where (*L*, $\varepsilon_{GaN}$), ($l_{1}$, $\varepsilon_{InN}$), and ($l_{2}$, $\varepsilon_{w}$) denote the layer thickness and the dielectric constant of the LQW, RQW, and the barriers, respectively. Hence, the total polarization ($P_{\nu }$), is given by the sum of both the piezo and spontaneous polarizations, and it is expressed as follows:(12)$${\;P}_{\nu }=P_{\nu }^{sp}+P_{\nu }^{pz}$$
where ν denotes either GaN or InN, while w denotes InGaN alloy.

The second-order expression for the spontaneous polarization of random ternary group III-nitride alloys, denoted by x, is given in units of C/m^2^ [45].
(13)$$P_{{In}_{x}Ga_{1-x}N}^{sp}={xP}_{InN}^{sp}+\left(1-x\right)P_{GaN}^{sp}+0.037x\left(1-x\right)$$

The piezoelectric polarization of the binary compounds is formulated as follows [46]:(14)$$\left\{\begin{array}{c} P_{InN}^{pz}=1.373\varepsilon +7.559\varepsilon^{2}\\ P_{GaN}^{pz}=-0.918\varepsilon +9.541\varepsilon^{2} \end{array}\right.$$

The piezoelectric field polarization along the c-axis resulting from the mismatch between the well and barrier materials is provided by the following equation [47]:(15)$${\;P}_{InxGa_{1-x}N}^{pz}=e_{31}\left(\varepsilon_{xx}+\varepsilon_{yy}\right)+e_{33}\varepsilon_{zz}$$
where $e_{ij}$ and $\varepsilon_{ij}$ are the piezoelectric constants and the strain elements of (In, Ga)N materials, respectively. These can be expressed analytically as follows:(16)$$\varepsilon_{xx}=\varepsilon_{yy}=\varepsilon =\frac{a_{b}-a_{w}\left(x\right)}{a_{w}\left(x\right)}$$
(17)$$\varepsilon_{zz}=-2\frac{c_{13}}{c_{33}}\varepsilon_{xx}\;$$
where $a_{b}$ and $a_{w}$ are the lattice parameters of the wells and the barriers, respectively, while $C_{13}$ and $C_{33}$ are the corresponding elastic constants. Hence, the revised expression for the piezoelectric polarization is presented as follows [48]:(18)$$P_{InxGa_{1-x}N}^{pz}=2\left(\frac{a^{b}-a^{w}}{a^{w}}\right)\left(e_{31}-e_{33}\frac{c_{13}}{c_{33}}\right)$$

### 2.3. Parameters Influenced by Pressure and Strain

The strain effects resulting from the mismatch in lattice constants between the LQW and RQW, as well as the barrier materials, can be taken into account by considering alterations in quantum confinement for both holes and electrons.

The strain parameter, *ε*, is derived from the lattices of InN and GaN. This can be given as follows [49]:(19)$$a_{InxGa_{1-x}N}\left(p\right)=xa_{InN}\left(p\right)+\left(1-x\right)a_{GaN}\left(p\right)$$

As a result, the strain parameter is determined as follows:(20)$$\varepsilon \left(p\right)=\frac{a^{b}\left(p\right)-a^{w}\left(p\right)}{a^{w}\left(p\right)}$$

As indicated in Ref. [50], the pressure-dependent behaviors of the lattice constants, well widths, and barrier widths are provided as follows:(21)$$L(p)=L(0)\left[1-\left(S_{11}^{GaN}+2S_{12}^{GaN}\right)p\right]$$
(22)$$l_{1}(p)=l_{1}(0)\left[1-\left(S_{11}^{InN}+2S_{12}^{InN}\right)p\right]$$
(23)$$l_{2}(p)=l_{2}(0)\left[1-\left(S_{11}^{w}+2S_{12}^{w}\right)p\right]$$
where $L\left(0\right),\;l_{1}(0)$, and $l_{2}(0)$ represent the lattice constant, the width of the barrier, and the widths of the LQW and RQW without the influence of pressure, respectively. Meanwhile, $S_{11}$ and $S_{12}$ correspond to the compliance constants expressed in terms of the elastic constants, $C_{ij}$, for GaN, InN, and InGaN materials, as follows [51]:(24)$$\left\{\begin{array}{c} S_{11}^{\nu }\left(p\right)=\frac{c_{11}^{\nu }\left(p\right)+c_{12}^{\nu }\left(p\right)}{\left(c_{11}^{\nu }\left(p\right)-c_{12}^{\nu }\left(p\right)\right)\left(c_{11}^{\nu }\left(p\right)+2c_{12}^{\nu }\left(p\right)\right)}\\ S_{11}^{\nu }\left(p\right)=\frac{-c_{12}^{\nu }(p)}{\left(c_{11}^{\nu }\left(p\right)-c_{12}^{\nu }\left(p\right)\right)\left(c_{11}^{\nu }\left(p\right)+2c_{12}^{\nu }\left(p\right)\right)} \end{array}\right.$$

Additionally, for GaN and InN materials, the pressure-dependent bandgap is given as follows [52]:(25)$${\;E}_{g}\left(\nu ,p\right)=E_{g}\left(\nu ,p=0\right)+\gamma \left(\nu \right)p+\delta \left(\nu \right)p^{2}$$
where ν represents either GaN material or InN material, and $E_{g}(\nu ,p=0)$ denotes the bandgap energy of the ν material at zero pressure. The correlation between the bandgap and the performance of the solar cells is explicitly addressed through the manipulation of the indium concentration in the material. The bandgap of the ${In}_{x}{Ga}_{1-x}N$ material can be derived using linear interpolation between InN and GaN, adjusted by the bowing parameter, as described in [53]:(26)$$E_{g}\left({In}_{x}Ga_{1-x}N,p\right)=xE_{g}\left(InN,p\right)+\left(1-x\right)E_{g}\left(GaN,p\right)-\lambda x(1-x)$$
where λ is the bowing parameter that considers the nonlinearity of the bandgap with respect to the indium composition; in this study, we took a value of λ=1.43 eV.

$m_{i}^{*}\;(x,p,z)$ and $\varepsilon^{*}(x,p,z)$ represent the effective mass and relative dielectric constant, respectively. In the case of (In, Ga)N, these parameters are expressed as linear combinations of the corresponding values for InN and GaN. Within the regions of the SGQW quantum structure, their definitions are as follows [54]:(27)$$m_{i}^{*}\left(x,p,z\right)=\left\{\begin{array}{c} m_{GaN}^{*}\left(p\right)0\;\leq z\leq L\\ m_{InN}^{*}\left(p\right)L\;\leq z\leq L+l_{1}\\ xm_{InN}^{*}\left(p\right)+\left(1-x\right)m_{GaN}^{*}\left(p\right)L+l_{1}\;\leq z\leq L+l_{1}+l_{2}\\ m_{GaN}^{*}\left(p\right)L+l_{1}+l_{2}\leq z\leq 2L+l_{1}+l_{2} \end{array}\;\right.$$
where $m_{i}^{*}(x,p,z)$ is the pressure-dependent effective mass; according to $\vec{k}\cdot \vec{p}$ theory, it is given by the following expression [50]:(28)$$m_{i}^{*}\left(x,p,z\right)=\frac{m_{0}}{1+\frac{c_{i}^{j}}{E_{g}^{j}(p)}}$$
where *m*_0_ is the free electron mass and $c_{i}^{j}$ is the energy-related momentum matrix element obtained by the previous equation without pressure at P=0, with $m^{*}(0)=\left[0.1x+\left(1-x\right)0.19\right]m_{0}$. The numerical values of those parameters are C(GaN)=14.7 eV and C(InN)=15.50 eV [34]. Furthermore, for the heavy holes, we employed effective mass values that are independent of pressure. The hydrostatic-pressure-dependent static dielectric constant of the ${In}_{x}{Ga}_{1-x}N$ material is derived through linear interpolation between the relevant values for InN and GaN. Given that ν represents either InN or GaN, the procedure is as follows [55]:(29)$$\varepsilon^{*}\left(x,p,z\right)=\varepsilon_{{In}_{x}Ga_{1-x}N}\left(p\right)=x\varepsilon_{InN}\left(p\right)+\left(1-x\right)\varepsilon_{GaN}\left(p\right)$$
with
(30)$$\varepsilon \left(\nu ,p\right)=\varepsilon_{\infty }\left(\nu ,p\right){\left[\frac{\omega_{Lo}\left(\nu ,p\right)}{\omega_{To}\left(\nu ,p\right)}\right]}^{2}$$
where
(31)$$\omega_{k}\left(\nu ,p\right)=\omega_{k}\left(\nu ,0\right)\;exp\left[\frac{\gamma_{k}\left(\nu \right)p}{B_{0}\left(\nu \right)}\right]$$
and k≡(Lo;To).
(32)$$\varepsilon_{\infty }\left(\nu ,p\right)=1+\left(\varepsilon_{\infty }\left(\nu ,0\right)-1\right)exp\left[\frac{-5}{3B_{0}\left(\nu \right)}(0.9-f_{i}\left(\nu \right)p)\right]$$

We chose to use the finite element method (FEM) to solve the Schrödinger equation of the studied system. This numerical technique subdivides complex physical problems into smaller and more manageable elements, using mathematical principles to approximate solutions within each segment. FEM, widely used in the fields of engineering and sciences, excels in solving a wide range of complex challenges in engineering, physics, and other areas due to its recognized versatility. Additionally, FEM demonstrates a talent for maintaining both accuracy and adaptability when modeling irregular geometries and material properties, making it invaluable for solving real-world practical problems [56]. In this study, the presence of a donor impurity in the structure makes the Schrödinger equation unsolvable through conventional analytical means. Therefore, we use FEM with a one-dimensional mesh (computational grid) consisting of 3N + 1 points, where N is set to 50. This approach provides accuracy for the ground and excited states of quantum-well (QW) systems. However, for more complex or higher-energy systems, the accuracy of the FEM solution may decrease, requiring a finer mesh and higher-order basis functions. Accuracy depends on factors such as the complexity of the problem, choices of numerical parameters, and computing resources, unlike conventional methods such as perturbative and variational techniques. It is important to note that, for the determination of energy levels and their corresponding wave functions, we consider the following boundary conditions [23,38,57,58,59]:(33)$${\left[\underset{n}{\to }.\underset{\nabla }{\to }\left(\frac{\psi }{m_{e,GaN}^{*}}\right)\right]}_{barrier}={\left[\underset{n}{\to }.\underset{\nabla }{\to }\left(\frac{\psi }{m_{e,InGaN}^{*}}\right)\right]}_{well}$$

The studied system utilizes a mesh grid with 4N + 1 points, where N is a fixed parameter. The discretization of each layer in the system involves different step sizes. Specifically, the step size for the barriers is denoted as $h_{b}=L/N$, while for the regions within the well it is expressed as $h_{w}=l/N$. Consequently, for k values ranging from 0 to N, the corresponding mesh nodes for a single quantum well (QW) can be determined as follows: the left barrier is positioned at $z_{j}=k\;*\;h_{b}$, the well region is located at $z_{j}=L+k\;*\;h_{w}$, and the right barrier is situated at $z_{j}=L+l+k\;*\;h_{b}$. Using the finite element method (FEM), we computed the first and second derivative wave functions [38].
(34)$${\left(\frac{\partial ²\psi (z)}{\partial z²}\right)}_{z_{k}}=\frac{\psi_{k+1}-2\psi_{k}+\psi_{k-1}}{{\left(z_{k+1}-z_{k}\right)}^{2}}$$
(35)$${\left(\frac{\partial \psi \left(z\right)}{\partial z}\right)}_{z_{k}}=\frac{\psi_{k+1}-\psi_{k}}{z_{k+1}-z_{k}}$$

Supposing that $h_{b}=z_{k+1}-z_{k},$ Equation (2) becomes
(36)$$\left(\frac{-\hbar^{2}}{2m_{e,hh}^{*}}\right)\left[\frac{\psi_{k-1}-2\psi_{k}+\psi_{k+1}}{{\left(h_{b}\right)}^{2}}\right]+V_{0}^{e,hh}\psi_{k}=E\psi_{k}$$

Assuming that $\Omega =\frac{-\hbar^{2}}{2m^{*}{h_{b}}^{2}},$ the same equation above becomes
(37)$$\Omega \left[\psi_{k-1}+\psi_{k+1}+\left(\frac{V_{0}^{e,hh}}{\Omega }-2\right)\psi_{k}\right]=E\psi_{k}$$

The matrix that furnishes the energy levels and corresponding wave functions in the specified region, namely, the barrier region, can be expressed as follows:(38)$$M_{Barrier}=\left(\begin{array}{cccccc} 0 & 0 & 0 & 0 & 0 & 0\\ \Omega & \left(V_{0}^{e,hh}-2\Omega \right) & \Omega & 0 & 0 & 0\\ 0 & \Omega & \left(V_{0}^{e,hh}-2\Omega \right) & \Omega & 0 & 0\\ 0 & 0 & 0 & 0 & \ddots & \vdots \\ 0 & 0 & 0 & 0 & 0 & \cdots \\ 0 & 0 & 0 & 0 & 0 & 0 \end{array}\right)$$

Comparable procedures can be employed to derive the matrix responsible for determining the energy levels and associated wave functions in the remaining regions (e.g., the well) by eliminating the potential ($V_{0}=0$), which is non-zero in any of the barrier regions. The system’s matrix is derived by combining the three computed matrices (left barrier, well, and right barrier). The numerical solutions for these matrices were implemented using the “Python programming language”, incorporating libraries like NumPy, SciPy, Math, Matplotlib, and others

### 2.4. Photonic and Electrical Characteristics of the Solar Cell

To numerically calculate the efficiency in the photoelectric conversion process for studying SGQW-IBSCs, it is necessary to adhere to the following principles outlined by Luque and Marti [60]: the solar cells must be of sufficient thickness to ensure complete photon absorption, non-radiative transitions are disregarded, and carrier mobility should be adequately high. With the aforementioned definitions of assumptions and concepts, we will present certain attributes of SGQW-IBSCs’ performance in the subsequent section. Additionally, the most crucial parameters in solar cell investigations encompass photon current density, open-circuit voltage and, ultimately, efficiency. In the context of full-concentration sunlight, both the number of photons absorbed by the solar cells and the number emitted from them determine the density of the photons generated. One of the primary physical parameters of a solar cell is the short-circuit current ($j_{sc}$), which can be formulated as follows [61]:(39)$$\frac{j_{sc}}{q}=\left[F\left(E_{13},\infty ,T_{s},0\right)-F\left(E_{13},\infty ,T_{c},\mu_{cV}\right)\right]+\left[F\left(E_{23},E_{12},T_{s},0\right)-F\left(E_{23},E_{12},T_{c},\mu_{cI}\right)\right]$$
where $T_{S}$ and $T_{C}$ represent the surface temperature of the Sun and the solar cell, respectively; q denotes the elementary charge, while $\mu_{cV}$ and $\mu_{cI}$ stand for the chemical potential differences between the conduction band (CB) and valence band (VB) and between the intermediate band (IB) and CB, respectively. $E_{13},\;E_{12},$ and $E_{23}$ are determined by solving the Schrödinger equation for the system. Based on the Roosbroeck–Shockley formula, the flux of photons (*F*) leaving an object at temperature *T* can be expressed as follows [62]:(40)$$F\left(u,v,T,\mu \right)=\frac{2\pi }{h^{3}c^{2}}\int_{u}^{v}\frac{E^{2}dE}{e^{\frac{\left(E-\mu \right)}{K_{B}T}}-1}$$
where u and v represent the lower and upper energy limits of the photon flux for the respective transitions, T denotes the temperature, h stands for Planck’s constant, C is the speed of light in a vacuum, $K_{B}$ is Boltzmann’s constant, and U signifies the chemical potential of the transition. On the other hand, for a p-i-n solar cell, the output voltage $V_{OC}$ of an IBSC can be expressed as follows [17]:(41)$$V_{oc}=\mu_{cV}=\mu_{cI}+{\mu \;}_{IV}$$
where $\mu_{cV}$ and $\mu_{IV}$ are given by the following expression [62]:(42a)$$\mu_{cV}=E_{23}+0.5∆e-E_{c}+E_{FC}$$
(42b)$$\mu_{IV}=E_{12}+0.5∆e-E_{FV}+E_{V}+V_{0h}+E_{h}^{1}$$
where ∆e is the width of the IB for the electron.

The quasi-Fermi levels $E_{FC}$ and $E_{FV}$ of the CB and VB, respectively, can be expressed as follows:(43)$$E_{c}-E_{Fc}=kT\ln\left(\frac{N_{c}}{n}\right)$$
(44)$$E_{FV}-E_{V}=kT\ln\left(\frac{N_{V}}{p}\right)$$
where $N_{C}$ and $N_{V}$ represent the effective densities of states in the conduction band (CB) and valence band (VB), respectively. Meanwhile, n and p denote the electron and hole concentrations, respectively, and they are given as follows [63]:(45)$$n=N_{c}\;exp\left[-\frac{Q∆{E_{g}}^{\Gamma }(x,P)}{K_{B}T}\right]$$
(46)$$p=N_{V}\;exp\left[-\frac{\left(1-Q)∆{E_{g}}^{\Gamma }(x,P\right)}{K_{B}T}\right]$$

The effective densities of states in the CB and VB ($N_{C}$ and $N_{V}$, respectively) are expressed as follows:(47)Nc=Nc* T32 
(48)NV=NV* T32

In the aim of improving the photovoltaic conversion efficiency in our study, we did not solely consider the optimal case of FF=1. Instead, we recognize that the fill factor (FF) is typically a function of the open-circuit voltage, $v_{OC}$, expressed as $V_{OC}/(KT/q)$, and is formulated as follows [64]:(49)$$FF=\frac{v_{OC}-Ln\left(v_{OC}+0.72\right)}{\;1+v_{OC}}$$

As a result, the efficiency of the QW-IBSC can be derived in the general case by utilizing the output voltage and photocurrent density, as follows [65]:(50)$$\eta =\frac{V_{oc}\cdot J_{sc}\cdot FF}{P_{in}}$$

It should be noted that ($P_{in}=\sigma T_{s}^{4}$) is the incident power coming from the Sun per unit of area and $\sigma =5.67\cdot 10^{-8}\;Wm^{-2}k^{-4}$ is the Stefan–Boltzmann constant.

## 3. Results and Discussion

The physical parameters used for numerical computations in our study of wurtzite GaN and InN are as follows: The bandgap ($E_{g}$) at zero pressure is 3.42 eV for GaN and 0.72 eV for InN [57]. The pressure-dependent parameters include γ (meV/GPa)=40 for GaN and 16 for InN, and $\delta \;(meV/GPa^{2})=-0.38$ for GaN and −0.02 for InN. The elastic constants are $C_{11}\left(GPa\right)=293$ for GaN and 187 for InN, and $C_{12}\left(GPa\right)=159$ for GaN and 125 for InN. The longitudinal and transverse optical phonon frequencies ($\omega_{L0}$ and $\omega_{T0}$, respectively) are $731.51\;cm^{-1}$ and $525.56\;cm^{-1}$ for GaN, and $621.53\;cm^{-1}$ and $487.67\;cm^{-1}$ for InN, respectively. The adiabatic longitudinal and transverse optical effective charges ($A_{L0}$ and $A_{T0}$, respectively) are 1.08 and 1.47 for GaN, and 1.27 and 1.52 for InN, respectively. The high-frequency dielectric constants $\varepsilon_{\infty }(P=0)$ are 4.942 for GaN and 6.723 for InN. The bulk modulus $B_{0}$ is 190 GPa for GaN and 136 GPa for InN. The piezoelectric constants $f_{i}$ are 0.1365 for GaN and 0.1406 for InN. The effective electron masses $\left(m_{e}^{*}/m_{0}\right)$ are $0.193\;m_{0}$ for GaN and $0.1\;m_{0}$ for InN. The effective hole masses $\left(m_{h}^{*}/m_{0}\right)$ are $0.810\;m_{0}$ for GaN and $0.1835\;m_{0}$ for InN. The low-frequency dielectric constants $\varepsilon (\varepsilon_{0})$ are 9.68 for GaN and 11.16 for InN [17,66,67,68]. In this study, our calculations were confined to scenarios where a second IB was not observed. Consequently, we solely considered the influence of a single IB. This research incorporates the consideration of both atmospheric and hydrostatic pressures. Specifically, under conditions of zero hydrostatic pressure, only atmospheric pressure was employed. To prevent any distortion of the material’s structure, we limited our analysis to cases corresponding to hydrostatic pressure values within the range of 0→ 30 GPa. Additionally, to streamline our computations in this numerical investigation, we employed effective atomic units. The effective Rydberg constant, $R_{b}^{*}=\left(\frac{m_{b}^{*}e^{3}}{2(4\pi \varepsilon_{b}^{*}{\hbar )}^{2}}\approx 29.81\;meV\right)$, was adopted as the energy unit, and the effective Bohr radius, $a_{b}^{*}=\left(\frac{4\pi \varepsilon_{b}^{*}\hbar^{2}}{m_{b}^{*}e^{2}}\approx 2.29\;nm\right)$, served as the unit of length, along with a dimensionless parameter that accounted for the BEF effect. These choices are directly tied to the optical energy transitions, which are closely linked to the key variables of photovoltaic conversion. Our calculations were conducted initially for an impurity located at $z_{0}=L+\frac{l_{1\left(0\right)}}{2}$ with a fixed InN layer thickness, taken as $l_{1}\left(P=0\right)=2a_{b}^{*}$, at room temperature (T=300 K). With all of the parameters and methodologies outlined, we can now initiate the discussion of the obtained results.

Figure 2 depicts the variation in the energies of both electrons and holes with respect to the In concentration for different barrier width values (Figure 2a,b), RQW (InGaN layer) width values (Figure 2c,d), and hydrostatic pressures (Figure 2e,f). Despite the constraints imposed, it is clear that the energy levels of both electrons and holes decrease with an increase in indium composition. This decline can be attributed to the reduction in quantum confinement arising from the inclusion of indium. Moreover, Figure 2a,b demonstrate that this energy reduction is more significant with an increase in barrier size (L). However, this effect is more pronounced for electron energy compared to hole energy.

This is due to a decrease in spatial confinement within the conduction band compared to the confinement within the valence band. Similarly, in Figure 2c,d, it is worth noting that widening the RQW leads to a decrease in energy, although this decline is minimal for lower indium compositions and more significant for higher indium ratios, especially for holes. This is because increasing the RQW size reduces quantum confinement, lowering the energies of both electrons and holes as they become less restricted within the well region. However, as shown in Figure 2e,f, increasing the hydrostatic pressure results in a reduction in electron energy. This decrease in energy is more pronounced at lower indium concentrations and becomes less steep as the indium composition increases. In contrast, it results in a gradual increase in hole energy. This occurrence can be attributed to structural deformation induced by increasing pressure, impacting the energies of both electrons and holes. As the pressure rises, the electron energies decrease due to heightened constraints on their ability to escape and penetrate barrier regions. This restriction causes the electrons’ wave functions to relax, contributing to the observed decline in their energies. On the other hand, for holes, the increase in pressure results in greater confinement, leading to an increase in their associated energy.

Figure 3 displays the optical transition energy ($E_{12}$ and $E_{23}$) changes with respect to the In concentration, considering the impact of barrier width (Figure 3a), the LQW’s width (Figure 3b), and hydrostatic pressure (Figure 3c). In Figure 3a, the alterations in optical transition energies $E_{12}\;(1s\to 2p)$ and $E_{23}\;(2p\to 3s)$ are presented as a function of the In content. The data correspond to a hydrostatic pressure of P=10 GPa and $l_{2}\left(0\right)=1.0a_{b}^{*}$. These results are showcased for four distinct barrier width values, in the absence of pressure-induced changes L(0). The trend is evident: the transition energy ($E_{23}$) from the intermediate band to the conduction band shows a rapid and linear decline as the In content increases. $E_{23}$ demonstrates a reduction of approximately 86.6% as the In composition is increased from 35% to 74%. This outcome can be rationalized by considering the escalating BEF as x increases. Consequently, the shapes of both the conduction and valence bands tilt and deepen. This phenomenon primarily arises from the QCSE (quantum-confined Stark effect), leading to a redshift in the transition energy and a decrease in the transition probability. This shift can be attributed to the spatial separation between electrons and holes caused by the internal piezoelectric field. Moreover, this decrease exhibits minimal alteration upon adjusting the barrier width. Secondly, the interband transition energy from the hole level to the intermediate band ($E_{12}$) experiences a slight decline as the In content increases.

Interestingly, this energy remains unchanged with respect to the barrier width L(0). Indeed, $E_{12}$ demonstrates a decrease of approximately 6.76% within the same range of In content. This observation is rationalized by the fact that the absolute value of the BEF increases in tandem with the x-component. This increase can be attributed to alterations in the biaxial deformation pressure of the InGaN layer as x escalates. These changes exert an influence on the piezoelectric polarization. Notably, the strength of $F_{GaN}$ and $F_{InGaN}$ increases with the increase in piezoelectric polarization, as outlined in Equation (11). Figure 3b demonstrates the same transition energies, $E_{12}$ and $E_{23}$, in relation to the indium fraction across four different values of $l_{2}\left(0\right)$, while keeping a constant barrier width of $L(0)=4.0a_{b}^{*}$. Similar trends can be observed compared to Figure 3a. A notable 85.57% reduction can be seen for $E_{23}$. However, distinct behaviors emerge for $E_{12}$. In the case of low In contents (0.35<x<0.54), a slight decrease in $E_{12}$ energy is observed, irrespective of $l_{2}(0)$. For higher In contents (0.54<x<0.74), the decrease is more pronounced and closely tied to $l_{2}(0)$. For instance, this reduction amounts to 3.45% and 8.30% for $l_{2}(0)$ values of 0.5 and $2.5a_{b}^{*}$, respectively. This outcome can be attributed to the influence of $l_{2}(0)$, particularly for In contents exceeding 0.54. Notably, this effect surpasses the influence of the BEF within this range of In content. Consequently, the utilization of a GaN-based semi-graded quantum well structure becomes relevant to mitigate the impact of the potent built-in electric field arising from piezoelectric and spontaneous polarizations in WZ GaN-based QWs. It is crucial to note that the transition energy $E_{13}$, representing the host bandgap, is not displayed, as it remains unaffected by the studied factors. However, Figure 3c displays the optical transitions $E_{12},\;{\;E}_{23}$, and $E_{13}$ with respect to In content, maintaining fixed values for barrier width ($L\left(0\right)=6a_{b}^{*}$) and LQW width ($l_{2}\left(0\right)=1.0a_{b}^{*}$) while examining four distinct pressure levels. Notably, the host bandgap $E_{13}$, incorporating hole levels and responding to increased pressure, adheres to the relationship portrayed in Equation (25). Importantly, $E_{13}$ remains unaffected by changes in indium concentration. The choice of barrier material, such as GaN, significantly shapes the behavior of the host bandgap $E_{13}$, underscoring the pivotal role of material selection in enhancing device functionality. The intersubband ground-state transition (IB–CB) between the intermediate and conduction bands, $E_{23}$, experiences a reduction with increasing In composition. As the indium concentration goes from 0.35 to 0.74, a steep decline of about 83.22% at P=30 GPa and approximately 88.44% at P=0 GPa is noticeable. This drop becomes less pronounced with increasing pressure. Notably, at lower concentrations, pressure yields a positive effect on the $E_{23}$ transition energy, while it exerts a negative impact at higher concentrations. This phenomenon can be explained by the gradual contraction of the quantum well’s size as pressure increases, resulting in an intensified influence of quantum confinement. Additionally, as the energy increases, the band offset also experiences an increase. The interband transition energy $E_{12}$, spanning from the hole level to the intermediate band, exhibits a marginal decline with increasing In content, regardless of the pressure setting. Within the same range of x, the reduction in transition energy is 6.52% at P=0 GPa and 3.97% at P=30 GPa. Furthermore, for a given In concentration, the transition energy $E_{12}$ increases with pressure. This phenomenon can be rationalized by the fact that as pressure rises, there is a concurrent increase in spontaneous and piezoelectric polarization, consequently elevating the electric fields of $F_{GaN},\;\;F_{InGaN}$, and $F_{InN}$. These BEFs induce a separation of electrons and holes in the opposite direction, leading to their wave functions’ overlap. As a result, the electron and hole energies increase, leading to an observable enhancement in the $E_{12}$ transition energy. Figure 4 illustrates the variation in the IB width versus In composition at room temperature, focusing on the on-center impurity case for a fixed $l_{1}(0)$ = 2.0$a_{b}^{*}$ and considering the impacts of barrier/well thicknesses and pressure. Regardless of the pressure and InN layer and InGaN layer thicknesses, Figure 4a indicates a nonlinear decrease in IB width as a function of In composition. Notably, for a given indium molar fraction, the IB width diminishes as the barrier width increases. Additionally, within the range of studied chemical compositions, the IB width exhibits a reduction of approximately 53.88% and 54.96% for L(0) values of 2.0 and 8.0$a_{b}^{*}$, respectively. This suggests that the IB becomes narrower with higher indium content compared to lower indium content. It is indispensable to consider that our calculations are valid only in the absence of a second intermediate band; thus, our analysis focuses solely on the impact of one intermediate band. In Figure 4b, a prominent feature is the consistent reduction in IB width as a function of increasing In composition. Irrespective of the width of the InGaN layer (third layer) of length $l_{2}(0)$, it is evident that elevating the In composition results in a narrower IB. The influence of third layer’s width remains minimal within chemical compositions ranging from 35 to 53.24%, where the IB width experiences a linear decrease regardless of the specific width of the third layer. However, in the range from 53.24 to 74%, except for a third layer width of 0.5 nm, this reduction becomes nonlinear, with the third layer’s width exerting a substantial impact. This underscores the significance of employing an SGQW structure to mitigate the impact of the robust built-in electric field (BEF) [63,64]. Additionally, it is important to note that the built-in electric field of the ${\mathrm{In}}_{x}{\mathrm{Ga}}_{1-x}\mathrm{NQW}$ is estimated to have a magnitude of MV/cm. Figure 4c distinctly illustrates the substantial influence of pressure on the width of the IB. Particularly, as pressure rises, the IB width expands. Interestingly, regardless of the pressure value, variations in the IB width exhibit a consistent behavior: a nonlinear decrease relative to the chemical composition. This decline manifests in a decrease rate of approximately 56.98% at pressures of P=0 GPa and 51.77% at pressures of P=30 GPa.

Having delved into the concepts of energy encompassed in the investigated system, let us now redirect our attention to the fundamental aspects that dictate photovoltaic conversion. These critical factors encompass the open-circuit voltage ($V_{oc}$), the short-circuit current density ($J_{sc}$), and the fill factor (FF). It is noteworthy that the FF is intricately related to $V_{oc}$. As a result, these latter two parameters assume pivotal roles in shaping the device’s overall efficiency.

Figure 5 illustrates the alterations in the $V_{oc}$ concerning the In fraction at a temperature of T = 300 K, focusing on the on-center impurity case and employing a $l_{1}\left(0\right)=2a_{b}^{*}$. In Figure 5a, the variations in $V_{oc}$ are displayed for four distinct barrier widths, with a constant P = 10 GPa and $l_{2}\left(0\right)=1a_{b}^{*}$. The investigation reveals a nonlinear decrease in $V_{oc}$ as the In content increases. This decline in $V_{oc}$ is remarkably influenced by the presence of spontaneous and piezoelectric polarization, which intensifies with increasing chemical composition. Furthermore, the outcomes indicate that $V_{oc}$ experiences modest reductions as the barrier width increases. Specifically, the $V_{oc}$ decreased from 1.28 to 1.23 V, depicting a decay rate of approximately 3.9%, as L(0) increased from 2.0 nm to 8.0 nm. Furthermore, the $V_{oc}$ is closely tied to the difference in chemical potential between the valence band (VB) and conduction band (CB), which essentially represents the host material’s bandgap. As a result, changes in the barrier width have a negligible impact on $V_{oc}$. This indicates that the open-circuit voltage is minimally influenced by variations in L(0).

In Figure 5b, we depict the fluctuation in the $V_{oc}$ while maintaining constant parameters: P=10 GPa and $L\left(0\right)=4.0a_{b}^{*}$. The focus is on four distinct values of the width of the InGaN layer, denoted as $l_{2}(0)$. Evidently, $V_{oc}$ demonstrates a propensity to decrease as the In fraction increases. This apparent reduction in $V_{oc}$ can be attributed in part to the rise in defect density induced by higher indium fractions, as well as the substantial impact of spontaneous and piezoelectric polarization. Furthermore, this figure reveals the existence of a critical In fraction ($x_{c}=52.34\%$) that demarcates two distinct behaviors: For values of x less than $x_{c}$, it is evident that $V_{oc}$ experiences a linear decline, regardless of the specific width of the third layer. However, for x values greater than $x_{c}$, the influence of the InGaN layer becomes significant. Except for the curve corresponding to a third-layer width of $l^{2}\left(0\right)=0.5\;nm$, $V_{oc}$ exhibits nonlinear decay. As the chemical composition transitions from 0.35 to 0.74, the $V_{oc}$ demonstrates a decline rate of approximately 13.07% and 20.63% for $l_{2}(0)=0.5a_{b}^{*}$ and $l_{2}(0)=2.5a_{b}^{*}$, respectively. In Figure 5c, we present the variation in the $V_{oc}$ while maintaining fixed parameters, $L(0)=4.0a_{b}^{*}$ and $l_{2}(0)=1.0a_{b}^{*}$, for four distinct pressure values.

It is obvious that as the In concentration increases, $V_{oc}$ experiences a monotonic decrease. Additionally, a clear pattern emerges where the $V_{oc}$ increases in a consistent manner with increasing pressure, regardless of the chemical composition. Furthermore, $V_{oc}$ displays an almost linear increase from about 1.1 V at P=0 GPa to 1.6 V at P=30 GPa. This important increment is predominantly attributable to the increase in the bandgap resulting from increased hydrostatic pressure. This effect showcases a favorable impact on the overall performance of the device. However, it is essential to acknowledge that despite these enhancements, the values of $V_{oc}$ remain relatively weak due to the influence of spontaneous and piezoelectric polarizations.

Figure 5 shows the acquired outcomes of the photogenerated current density ($J_{sc}$) with respect to the influence of In composition. This examination was conducted at room temperature, incorporating on-center impurity and $l_{1}(0)=2\;nm$. Figure 6a depicts the variability in $J_{sc}$ under specific parameters: $P=10\;GPa,l^{2}\left(0\right)=1.0\;nm$, and four distinct barrier widths. It can be seen that the $J_{sc}$ displays an almost linear rise followed by a stabilization with the increase in chemical composition, while exhibiting only marginal sensitivity to the influence of L(0).

This behavior is rooted in the phenomenon whereby augmenting the indium fraction concurrently reduces the bandgap of ${\mathrm{In}}_{x}{\mathrm{Ga}}_{1-x}N$ and, therefore, the depth of the InGaN layer. Consequently, this reduction leads to lower transition energies, facilitating enhanced photon absorption. In simpler terms, the intrinsic (i) region absorbs more light energy and generates a higher number of carriers with the increase in the composition, subsequently promoting greater electron excitation within the material system. As emphasized throughout this study, it is imperative to note that our calculations are confined to scenarios where no second IB is observed. Moreover, due to the effects induced by spontaneous and piezoelectric polarization, the values of $J_{sc}$ remain relatively moderate.

Figure 5b elucidates the variability in $J_{sc}$ while maintaining fixed parameters: P=10 GPa and $L(0)=1.0a_{b}^{*}$. The exploration extends to four distinct values of the width of the InGaN layer, denoted as $l_{2}(0)$, in the absence of pressure effects. An increase in In composition yields a corresponding rise in $J_{sc}$, irrespective of the specific width of the InGaN layer. This enhancement in $J_{sc}$ as the In content increases can be attributed to an elevated generation rate and is rooted in the consequences of the amplified composition on the bandgap and the InGaN layer depth. This adjustment leads to a reduction in transition energies, thus facilitating heightened generation rates. This broader range of absorbed photon energies results in the creation of additional electron–hole pairs, thereby augmenting $J_{sc}$. Conversely, it has been demonstrated that the indium fraction reaches a critical value, $x_{c}=48\%$, above which the influence of the right quantum well (RQW) becomes distinctly evident. The narrower the width of the RQW, the greater its impact in minimizing the influence of the strong BEF. Hence, the interest in using quantum wells in semi-graded structures.

The influence of pressure on the $J_{sc}$ was investigated under fixed parameters: $l_{2}(0)=1.0a_{b}^{*}$ and $L(0)=4.0a_{b}^{*}$. As demonstrated in Figure 5c, an increase in In-content correlated with an augmentation in $J_{sc}$ across all pressure values. Moreover, the panel distinctly illustrates that, at a given concentration, increased pressure leads to a more substantial upswing in $J_{sc}$. Remarkably, this pressure-related impact is significantly pronounced for higher concentrations compared to lower concentrations. Consequently, increasing the pressure serves to enhance the $J_{sc}$. This behavior finds its explanation in the interplay between pressure and the energy of absorbed photons. With increasing pressure, the energy of absorbed photons also rises, owing to the concurrent augmentation in bandgap energy associated with pressure-induced effects. Figure 7 presents the numerical outcomes that we obtained regarding the photovoltaic conversion efficiency (η) plotted versus the In composition. These calculations were conducted under the condition of an on-center impurity scenario and at room temperature, with the parameter $l_{1}(0)=2.0a_{b}^{*}$. For all panels, and irrespective of the parameters under consideration, the photovoltaic conversion performance revealed a remarkable and significant finding: the existence of a critical In fraction value at which efficiency reaches its peak. This overarching trend is distinctly elucidated within the range of 0.5<x<0.6. Within this span, almost all optical transitions achieve their most favorable states, resulting in heightened absorption and an enhanced current density. These combined effects synergistically contribute to the augmentation of efficiency to its optimal level. In addition, the decrease in photovoltaic conversion performance beyond the critical indium compositions can be clarified by the deterioration in the quality of the InGaN layer. This deterioration can be attributed to the presence of spontaneous and piezoelectric polarizations, coupled with dislocation effects due to the rise in lattice mismatching between the layers. These challenges stem from the complexities encountered in achieving high-quality InGaN-based layers with indium compositions exceeding 50%, collectively contributing to the degradation of the device’s performance [35]. In Figure 7a, we delve into the variation in photovoltaic efficiency (η) for a specific pressure value at P=10 GPa and $l_{1}\left(0\right)=2l_{2}=2\;a_{b}^{*}$ across four distinct values of GaN layer thickness; η shows an ascent to a maximum point followed by a descent for all considered compositions. In addition, an improvement in the photovoltaic conversion efficiency was observed with decreasing GaN layer thickness, $L\left(0\right):8\to 2a_{b}^{*},$ due to the simultaneous significant improvement in both the $J_{sc}$ and $V_{oc}$. The efficiency improves from 28.63 to 31.22% with the reduction in the GaN layer thickness, marking a substantial enhancement, estimated at roughly 11.5%. 

This behavior is rooted in the phenomenon that the intermediate bands (IBs) exhibit heightened photon absorption in tiny GaN layers, stimulating electron excitation and generating a robust current density, thereby enhancing efficiency. It is imperative to acknowledge that the contribution of spontaneous and piezoelectric polarization leads to the manifestation of lower photovoltaic conversion efficiency. Similarly, as shown in Figure 7b, we examined the influence of InGaN layer thickness on the variation in photovoltaic efficiency (η) with a fixed value of pressure (P=10 GPa) for specific GaN layer and InN layer thicknesses ($L(0)=4a_{b}^{*}$ and $l_{1}\left(0\right)=2\;a_{b}^{*},$ respectively).

It is evident from the latter panel that the photovoltaic conversion efficiency improved as the thickness of the InGaN layer decreased. This enhancement was more pronounced in cases of higher indium compositions. The influence of the InGaN layer’s thickness closely resembles that of the GaN layer’s thickness, particularly with lower indium compositions. However, Figure 7c plots the changes in photovoltaic efficiency versus composition for four distinct values of pressure with fixed structures ($L(0)=4a_{b}^{*}$, and $l_{1}\left(0\right)=2l_{2}=2a_{b}^{*}$) at room temperature. It can be seen clearly that the influence of hydrostatic pressure on the photovoltaic conversion efficiency of the IBSC is evident. It markedly enhances the conversion performance of the IBSC, with a more pronounced effect observed for higher indium compositions as opposed to lower compositions. In a fixed design where $L\left(0\right)=4a_{b}^{*}$ and $l_{1}\left(0\right)=2=l_{2}\left(0\right)=2a_{b}^{*}$, and with a composition of 60% (x=0.6), raising the pressure from 0 to 30 GPa results in an efficiency enhancement from around 16 to 65.3%. This corresponds to an estimated increase of about 24.62%. This phenomenon can be elucidated by considering the augmentation of hydrostatic pressure, which amplifies the production of electrons transitioning from the valence band to the conduction band. This enhancement subsequently boosts the generated photocurrent, leading to an overall improvement in IBSC efficiency. Finally, it is imperative to emphasize that our findings have been subjected to rigorous comparison with data compiled from the scholarly literature, demonstrating substantial concurrence, particularly aligning well with studies conducted by highly proficient researchers in well-established laboratories in terms of theoretical photon-conversion performance [4,6,10].

## 4. Conclusions

In conclusion, this study aimed to achieve a comprehensive understanding of the performance of GaN/InN/InGaN/GaN-based p-i-n semi-graded intermediate-band solar cells (SG-IBSCs) under various internal and external parameters, with a specific focus on pressure, indium composition, and layer thicknesses. The calculations were conducted within the effective mass approximation, accounting for the influences of built-in electric fields arising from spontaneous and piezoelectric polarizations. The key findings of our investigation are as follows: (i) a noteworthy enhancement in efficiency was observed with reduced layer thickness, particularly notable for higher indium concentrations; (ii) the performance of the SG-IBSCs reached a peak efficiency of approximately 31% before declining for indium concentrations in the InGaN alloy exceeding 60% (x~0.6); (iii) SG-IBSC performance demonstrated an improvement correlated with the increase in hydrostatic pressure, reaching an efficiency of approximately 68% under a pressure of 30 GPa with specific device design considerations and indium compositions. We anticipate that these insights will augment our previous contributions and provide a significant advancement in the field of photovoltaic technology, specifically within the domain of III-N-based quantum wells (SG-IBSC).

## Figures and Tables

**Figure 1 nanomaterials-14-00104-f001:**
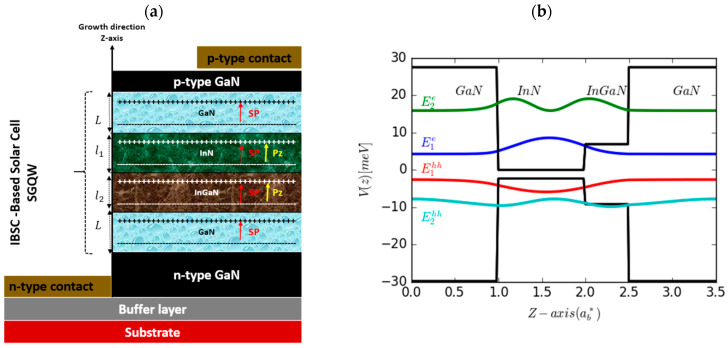
Panel (**a**) provides a detailed schematic (active region, contacts, and substrate), while panel (**b**) displays potential profiles, energy levels, and ground- and first-excited-state wave functions for electrons (CB) and holes (VB). $L=a_{b}^{*},\;l_{1}=2a_{b}^{*},\;l_{2}=0.5a_{b}^{*}$, with $a_{b}^{*}=2.29\;nm$ and x=20% (indium concentration).

**Figure 2 nanomaterials-14-00104-f002:**
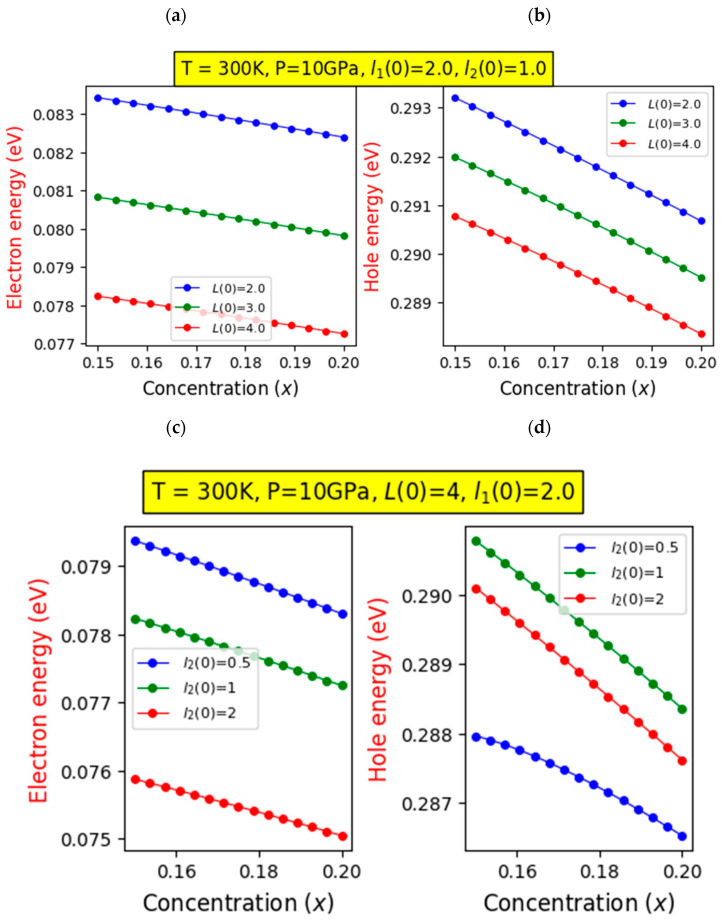
The energy levels of electrons and holes vary with changes in indium concentration (x), considering different thicknesses of the GaN layer (**a**,**b**), widths of the InGaN layer (RQW) (**c**,**d**), and adjustments to hydrostatic pressure (**e**,**f**).

**Figure 3 nanomaterials-14-00104-f003:**
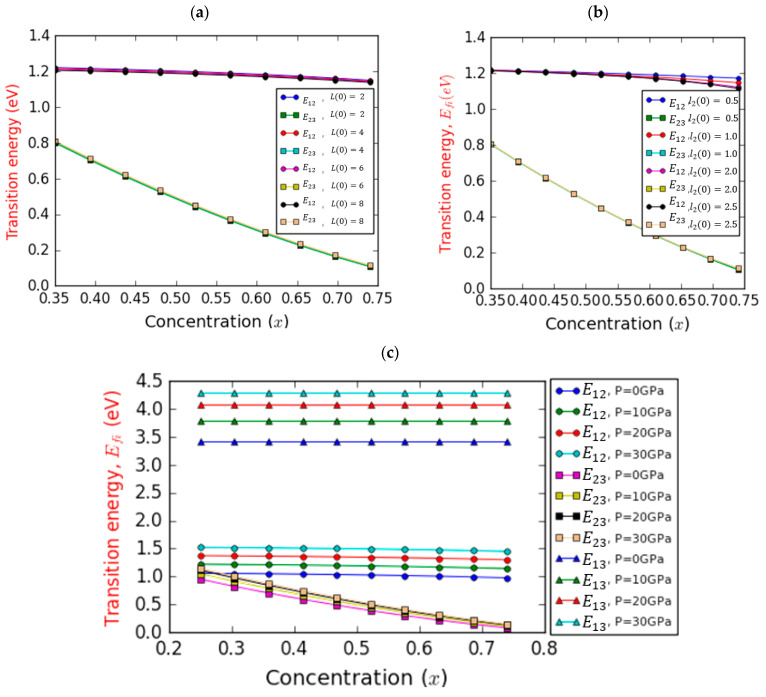
Variation in the optical transition energy as a function of the indium concentration (x) for different values of barrier width (**a**) with T = 300 K, $P=10\;\mathrm{GPa},\;l_{1}=2,\;l_{2}=1$, InGaN layer width (**b**) with T = 300 K, $P=10\;\mathrm{GPa},\;L=4,\;l_{1}=1$, and hydrostatic pressure (**c**) with T = 300 K, $L=6,\;l_{1}=2,\;l_{2}=1$.

**Figure 4 nanomaterials-14-00104-f004:**
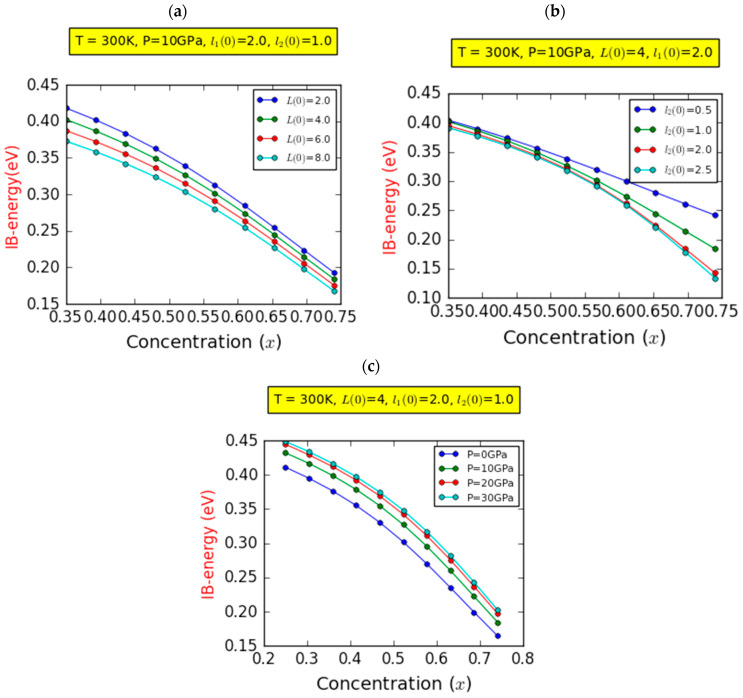
Variation in the intermediate band energy (IB) as a function of the indium concentration (x) for different values of barrier width (**a**), InGaN layer width (**b**), and hydrostatic pressure (**c**).

**Figure 5 nanomaterials-14-00104-f005:**
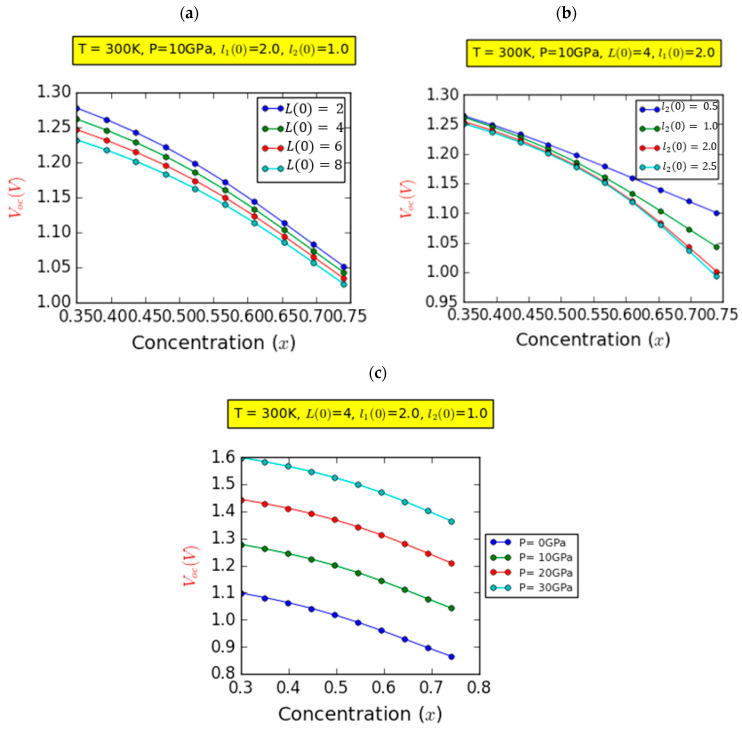
Variation in the open-circuit current ($V_{oc}$) as a function of the indium concentration (x) for different values of barrier width (**a**), InGaN layer width (**b**), and hydrostatic pressure (**c**).

**Figure 6 nanomaterials-14-00104-f006:**
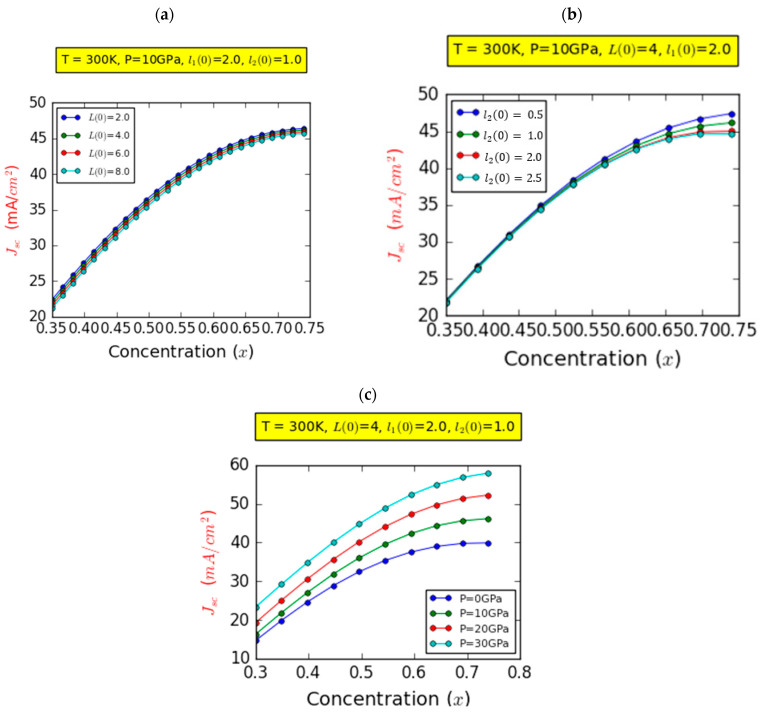
Variation in the short-circuit current ($J_{sc}$) as a function of the indium concentration (x) for different values of barrier width (**a**), InGaN layer width (**b**), and hydrostatic pressure (**c**).

**Figure 7 nanomaterials-14-00104-f007:**
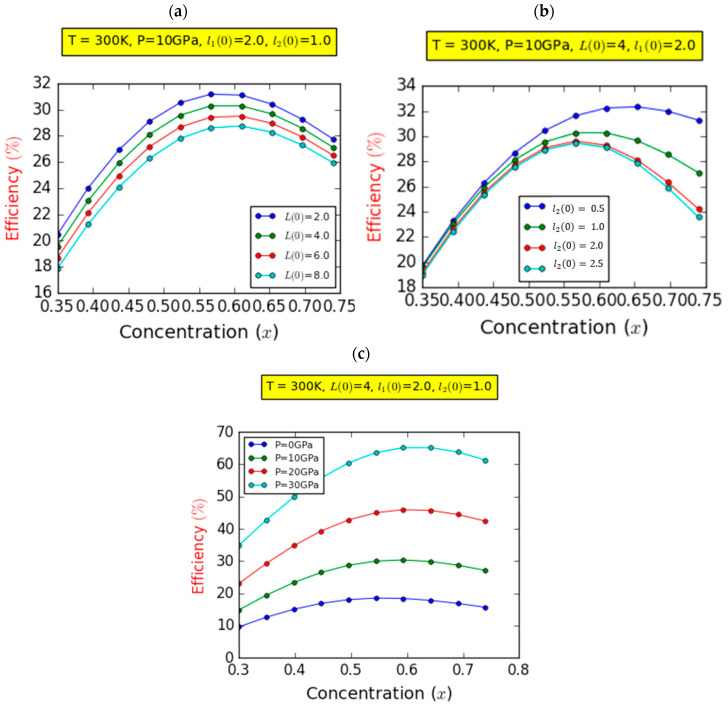
Variation in the efficiency (%) as a function of the In concentration for different values of barrier width (**a**), InGaN layer width (**b**), and hydrostatic pressure (**c**).

## Data Availability

Will be available upon request.

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
