# Peer review of "Efficiency of InN/InGaN/GaN Intermediate-Band Solar Cell under the Effects of Hydrostatic Pressure, In-Compositions, Built-in-Electric Field, Confinement, and Thickness"

_nanomaterials, 2024, doi:10.3390/nano14010104_

Round 1

Reviewer 1 Report (Previous Reviewer 2)

Comments and Suggestions for Authors

The authors responded to previous recommendations and revised the submission. What is interesting and requires publication is the crossing of the Schokley-Queisser limit by incorporating the quantum well structure. The revised version also mentions other methods of increasing the efficiency of perovskite cells like metallinc nanoparticles or quantum dots. The Stranski and Krastanov method of creating quantum dots is clarified. The article may be published after final proofreading.

Some comment on the manufacturing of a quantum well incorporated in cell structure would be beneficial.

Comments on the Quality of English Language

final proofreading is required 

Author Response

Thank you very much for your thorough review and constructive comments on our submission. We are grateful for your positive feedback, it is greatly appreciated. We sincerely appreciate your time and effort in reviewing our manuscript.

Reviewer 2 Report (New Reviewer)

Comments and Suggestions for Authors

In appreciated this work. So, this work aimed at improving the solar cells preformances. 

Comments on the Quality of English Language

In conclusion, this paper can accept for publication for minor revision. 

Author Response

Thank you very much for your thorough review and constructive comments on our submission. We are grateful for your positive feedback, it is greatly appreciated. We sincerely appreciate your time and effort in reviewing our manuscript.

Reviewer 3 Report (New Reviewer)

Comments and Suggestions for Authors

The article written by Abboudi et al. performed a computational study by focusing on optimizing the efficiency of III-nitrides-based Quantum Well Intermediate-Band Solar Cells. The work needs some major modifications.

1.       The paper's title could be more appealing; it should be revised.

2.       The abstract part needs to be more specific. Please revise the abstract and state clearly how this work is important for society and what the novelty of this work is, as well as key techniques/methods/results, etc.

3.       Please focus on the key things and add additional details in the introduction part. The introduction part needs some relevant literature for better understanding.

4.       Please add proper equations with references for each used calculation.

5.       The equations need to be better explained without references; please add references for all.

6.       What is the effect of layer thickness? Please add details.

7.       What is the effect of various interfaces? Please add details.

8.       Please add device optimization data.

9.       What is the energy alignment and bandgap of materials? Please add key details regarding this.

10.   The author should provide fill factor calculation details. Without calculating the fill factor, how can you estimate efficiency?

11.   Why Voc and Jsc increasing? In general, if one factor increases, the other decreases.

12.   please add calculation conditions and proper references in the methodology section.

13.   The impact of bandgap on the performance of SCs needs to be included.

14.   Please add the optical characteristics of these materials.

15.   Extensive English corrections are needed.

16.   The conclusion is too generic; please add key findings there. 

Comments on the Quality of English Language

Extensive editing of English language required

Author Response

Thank you very much for your thorough review and constructive comments on our submission. We are grateful for your positive feedback, it is greatly appreciated. We sincerely appreciate your time and effort in reviewing our manuscript.

Round 2

Reviewer 3 Report (New Reviewer)

Comments and Suggestions for Authors

Well revised-Accept

Comments on the Quality of English Language

 Extensive editing of English language required

This manuscript is a resubmission of an earlier submission. The following is a list of the peer review reports and author responses from that submission.

Round 1

Reviewer 1 Report

Comments and Suggestions for Authors

Dear authors,

This manuscript discusses the effects of In composition, built-in electric field, hydrostatic pressure, and layer thickness on Intermediate-band solar cell efficiency. In the GaN-based In(Ga)N epitaxial materials, the growth in the c (0001) axis often builds large misfit strain and pizeoelectric field. This work studies intermediate-band solar cells made of GaN/In(Ga)N quantum well whose energy bandgap can encompass 0.7~3.4 eV by varying In composition. It offers an extensive presentation of the theoretical basis as used and the calculation results it obtains, e.g. In composition effect, hydrostatic pressure effect, and quantum well geometry effect on energy band and solar cell performance. However, there are some issues of the structure design I have to clarify.

1. Intermediate-band solar cell is a new concept solar cell that embeds low-dimensional structures like quantum wells or quantum dots in the traditional bulk material solar cell (usually, single-junction one) with a goal to recycle sub-bandgap solar photons that cannot be absorbed in the bulk material. It is not the usual individual quantum well or quantum dot layer as used in light emission devices. The intermediate band is built from energy-state inter-layer coupling that broadens discrete energy levels in a low-dimensioanl structure to form a mini-band with electron-hole wavefunction overlap and recombination greatly varied (usually, a much longer recombination lifetime). In this band, both inter-band VB-IB trasition and intra-band IB-CB transition are allowed to recycle sub-bandgap photons. The spacing between quantum well or quantum dot layers must be very thin (e.g. 1~3 nm) to enable such state overlap. Besides, the layer number must be as many as possible to recycle sub-bandgap photons efficiently. Thirdly, GaN bulk has a bandgap of 3.4 eV which only absorbs very little part of solar light and the optimal intermediate-band banggap for the intermediate-band solar cell is 0.7 eV [Phys. Rev. Lett. 78, 5014 (1997)]. In this case, the InGaN/GaN system is not desired either from bandgap allocation or strain misfit accumulation. I suggest maybe a InGaN/InAlN system with less misfit strain is better.

2. As the simulation results in Figure 2c,2d,3b,4b and 5b show, the difference is only at high-In content region with a great pizeo-electric field effect. In experiment, since there is a great misfit strain between GaN and InN, it is difficult to directly grow InN layer on GaN surface but with few dislocations. So, the InGaN/GaN quantum structure design in Figure 1a must be modified with a lower In content offset (e.g. 0.2).

3. The hydrostatic pressure effect as presented shows a simulated pressure as high as 30 GPa. For solid-state system, such high pressure is usually obtained in a micro-region (e.g. in single quantum dot for quantum light application[Appl. Phys. Lett. 103, 252108 (2013)]). For solar cell application, it is difficult in experiment to build such high pressure in a large area.

Reviewer 2 Report

Comments and Suggestions for Authors

The paper deals with increasing the efficiency of solar cells by using intermediate levels in the forbidden gap of a semiconductor by immersing a quantum well in the thin-film structure of the cell. This is similar to the use of quantum dots that increase absorption and help manage photon energy.

The paper is worth publication. Some revision would be, however, of order. In the Introduction the other methods to surpass the Shockley-Queisser limit should be mentioned, as presented e.g., in Materials 2022, 15, 2254. https://doi.org/10.3390/ma15062254 . In particular, it would be very interesting to discuss a possibility of further improvement of the efficiency by simultaneous application of quantum well (or, alternatively, quantum dots) and plasmonic nano-metallic components, as both increase photon abosorption. It would be also beneficial to mention about other ways to increase the overall solar cell efficiency by non-optical effects, not related with photon absorption itself. As has been evidences both experimentally and theoretically, some internal electric-type effects are of signifficanace and can strongly increase overall efficiency (by even 40% of the relative increase) via reducing of photo-exciton binding energy and in this way accelerating exciton dissociation in perovskite thin layer cells highly increasing a photocurrent (Nano Energy 75 (2020) 104751). Summing up of both effect (increase of absorption dute to quantum well and reducing of excititon binding)  in such solar cells might be very interesting. 

In addition, in line 58, the method of selfassembling of quantum dots by Stranski and Krastanow technique is mentioned without any comment nor citation – this should be clarified (maybe other methods of quantum dot creation could be also mentioned). Similarly with regard to a quantum well, some additional comments on manufacturing of such structure would be beneficial.

Comments on the Quality of English Language

the paper is well written 

only final proofreading is recommended